# WO_3−x_/WS_2_ Nanocomposites for Fast-Response Room Temperature Gas Sensing

**DOI:** 10.3390/molecules30030566

**Published:** 2025-01-26

**Authors:** Svetlana S. Nalimova, Zamir V. Shomakhov, Oksana D. Zyryanova, Valeriy M. Kondratev, Cong Doan Bui, Sergey A. Gurin, Vyacheslav A. Moshnikov, Anton A. Zhilenkov

**Affiliations:** 1Micro- and Nanoelectronics Department, Saint Petersburg Electrotechnical University “LETI”, Professora Popova 5, 197022 Saint Petersburg, Russia; oksana.zyryanova.2002@gmail.com (O.D.Z.); congdoan6997@gmail.com (C.D.B.); vamoshnikov@mail.ru (V.A.M.); 2Institute of Artificial Intelligence and Digital Technologies, Kabardino-Balkarian State University, 360004 Nalchik, Russia; shozamir@yandex.ru; 3Moscow Center for Advanced Studies, Kulakova Str. 20, 123592 Moscow, Russia; kvm@mail.ru; 4Center for Nanotechnologies, Alferov University, Khlopina 8/3, 194021 Saint Petersburg, Russia; 5Department of Information and Measurement Equipment and Metrology, Penza State University, Krasnaya Street 40, 440026 Penza, Russia; teslananoel@rambler.ru; 6Department of Cyber-Phisical Systems, Saint Petersburg Marine Technical University “SMTU”, Leninsky Pr. 101, 198303 Saint Petersburg, Russia; zhilenkovanton@gmail.com

**Keywords:** semiconductor gas sensor, WO_3−x_/WS_2_ nanostructures, hydrothermal method, room temperature, alcohol vapor detection

## Abstract

Currently, semiconductor gas sensors are being actively studied and used in various fields, including ecology, industry, and medical diagnostics. One of the major challenges is to reduce their operating temperature to room temperature. To address this issue, sensor layers based on WO_3−x_/WS_2_ nanostructures synthesized by the hydrothermal method have been proposed. In this paper, the morphology of the material’s surface and its elemental composition were investigated, as well as the optical band gap. Additionally, changes in the resistance of the WO_3−x_/WS_2_ sensor layers under the influence of alcohol vapors at room temperature were analyzed. The results showed that the layers exhibited a significant response, with short response and recovery times. The achieved response value to 1000 ppm of isopropanol was 1.25, with a response time of 13 s and a recovery time of 12 s. The response to 1000 ppm of ethanol was 1.35, and the response and recovery times were 20 s. This indicates that these sensor layers have promising potential for various applications.

## 1. Introduction

Semiconductor gas sensors are widely used in various industries, including mining, metallurgy, chemistry, and fuel production, to detect toxic, flammable, and explosive gases. This helps ensure the safety of workers and the production process. Additionally, gas analytical devices can be used in medicine, environmental monitoring, and the food industry [1,2].

The operation of these sensors is based on the change in resistance when they interact with gaseous molecules [3,4,5,6]. Semiconductor gas sensors are highly sensitive and selective, with a fast response time and stable operating parameters [7,8,9,10]. They also have the advantages of compactness, simplicity of design, mechanical strength, and low cost, which make them widely available and prevalent [11,12,13]. However, for accurate and fast operation, there is an optimal temperature range of 200–500 °C [14,15], which can complicate the design and increase energy consumption. There is also a need to enhance the sensitivity and specificity of semiconductor sensors. For this purpose, new nanomaterials and nanostructures are being developed and studied [16,17,18,19,20,21,22]. Additionally, more complex devices are being created, for example, multisensor systems of the “electronic nose” type [23,24,25].

Various metal oxides, such as SnO_2_ [26], ZnO [27], TiO_2_ [28], WO_3_ [29], and Fe_2_O_3_ [30], are widely used to create chemiresistive gas sensors. Among these, tungsten oxide (WO_3_) stands out for its stability [31]. The production of oxide materials with a non-stoichiometric composition has been shown to improve the performance of sensors. Oxygen vacancies in these materials contribute to an increased sensor response, as several studies have demonstrated [32]. This is because oxygen vacancies act as sites for the adsorption of negatively charged oxygen molecules, which can react with target gas molecules [33].

Various methods are widely used to produce tungsten oxide, for example, thermal evaporation [34], pulsed laser deposition [35], hydrothermal method [36], chemical vapor deposition [37], etc. Moreover, the monoclinic WO_3−x_ non-stoichiometric phase can be easily achieved by various synthesis approaches, including both physical and chemical methods [32].

Studies have shown that the addition of various substances to tungsten oxide can improve its gas-sensing properties. As a result, composites such as SnSe_2_/WO_3_ [38], g-C_3_N_4_/WO_3_ [39], MWCNT/WO_3_ [40], and ZnO/WO_3_ [41] have been developed. Among these, WO_3_/WS_2_ has received particular attention [42,43,44]. Tungsten disulfide (WS_2_) has a large surface area and unique electronic properties, which have led to its use in various applications, including sensor layers in gas analytical devices [45,46], field-effect transistors [47], solar cells [48], LEDs [49], neuromorphic devices [50], and biosensors [51]. WS_2_ can be produced using various methods, such as mechanical and chemical exfoliation, chemical vapor deposition, hydrothermal synthesis, thermal decomposition, and magnetron sputtering [46,52]. In this way, nanostructures can have various shapes and sizes, including quantum dots, heterostructures, nanorods, nanoflowers, and nanosheets [52].

Chang et al. [53] showed that 2D WO_3_ nanosheets, produced by WS_2_ oxidation, can be used to create acetone sensors. The response to 50 ppm of acetone at 300 °C was 14.7. Additionally, the composite hierarchical structures of WS_2_/WO_3_ exhibited a response to H_2_, NH_3_, and NO_2_ at 150 °C [54]. Liang et al. [55] and Simon Patrick et al. [56] developed WS_2_/WO_3_ composites that respond to NO_2_ at room temperature. Thus, in most studies, the response was achieved at elevated temperatures, except in cases where NO_2_ was used as the detected gas. The development of composite sensor layers based on WO_3_, showing a response at room temperature, is an urgent task in modern gas sensors.

Volatile organic compounds released into the atmosphere as a result of industrial production pose a threat to the environment and human health. Among them, isopropyl alcohol C_3_H_8_O, used in medicine, chemical, and other industries, has narcotic and toxic effects on humans. Ethanol detection is of interest for assessing the condition of drivers in order to ensure safety on the road, as well as in the food industry.

A CuO-based sensor doped with Sn was proposed to assess the ethanol content in the air exhaled by drivers, the optimal operating temperature of which is 200 °C [57]. Gold-decorated SnO_2_ nanotubes can detect ethanol at 160 °C [58]. The yolk-shell Bi_2_MoO_6_ gas sensor showed a response to isopropanol at 270 °C [59]. MoO_3_ nanoflakes have also been developed, with an optimal detection temperature of 350 °C for ethanol and 200 °C for isopropanol [60]. In a number of studies, sensors based on tungsten oxide have been obtained, demonstrating high operating temperatures for response to alcohols, as shown in [61,62,63]. However, the problem of high operating temperatures in alcohol sensors has not yet been fully addressed in most work in this field.

The goal of this study is to synthesize WO_3−x_/WS_2_ composite nanostructures in order to develop a gas sensor capable of detecting alcohols at room temperature.

## 2. Results and Discussion

During the hydrothermal synthesis process, sodium tungstate decomposes:(1)$${Na}_{2}WO_{4}\to 2{Na}^{+}+WO_{4}^{2-}$$

Then, WO_3−x_ forms under acidic pH conditions:(2)$$WO_{4}^{2-}+2H^{+}\to WO_{3-x}+H_{2}O$$

The formation of WS_2_ occurs through the participation of L-cysteine, which breaks down in solution to release hydrogen sulfide (H_2_S). H_2_S then reacts with WO_3−x_ to create WS_2_. As a result of this hydrothermal synthesis process, WO_3−x_/WS_2_ composite materials are formed.

Analysis of SEM images (Figure 1a) showing the surface microstructure reveals that the composite materials consist mainly of 1D nanostructures up to 200 nm in length and approximately 10 nm in thickness. Additionally, some 2D nanosheets can be seen on the surface of the sensor layer. Based on optical microscopy data, the thickness of the sensor layer appears to be approximately 22–23 μm (Figure 1b).

Figure 2 shows the results of EDX spectroscopy for the synthesized WO_3−x_/WS_2_ sample, which allows us to characterize its elemental composition. This study confirms the presence of tungsten, oxygen, and sulfur in the sample.

The chemical composition of the sensor layer’s surface was analyzed using X-ray photoelectron spectroscopy. The survey spectrum of WO_3−x_/WS_2_ nanostructure, shown in Figure 3a, indicates that tungsten, sulfur, oxygen, and carbon are present in the sample. On the spectrum of the W 4f tungsten level, shown in Figure 3b, three peaks are observed, corresponding to binding energies of 33.6 eV, 35.1 eV, and 37.2 eV. The lowest-energy peak corresponds to the W 4f_5/2_ spectral line of W^4+^ tungsten in WS_2_ [64]. Two peaks with higher energies correspond to W 4f_7/2_ and W 4f_5/2_ for W^5+^ tungsten in WO_3−x_ [65]. The XPS spectrum for oxygen (Figure 3c) confirms the presence of the WO_3_ phase, as the binding energy of the O 1s line corresponds to that of metal oxides [66,67]. A peak with a lower intensity and binding energy of 530.9 eV most likely indicates the presence of oxygen vacancies [68]. The spectral 2p_1/2_ line of sulfur is difficult to distinguish in the survey spectrum, but its presence at a binding energy of 163.7 eV (Figure 3d) suggests that S^2−^ sulfides were found in the structure of the layer, most likely belonging to WS_2_ [67,69]. The minor component is probably related to the oxidized state of sulfur [70]. This peak corresponds to the W 4f_5/2_ peak for W^4+^ tungsten in WS_2_. Therefore, the synthesized sample contains a sensor layer in which WO_3−x_ and WS_2_ exist. The tungsten in the sample has various oxidation states, W^4+^ in WS_2_ and W^5+^ in WO_3−x_.

Based on the results of absorption spectroscopy, the optical band gap of the composite was estimated. A Tauc plot ${\left(\alpha *h\nu \right)}^{n}\left(h\nu \right)$ was constructed, where α is the absorption coefficient of the medium, and hν is the energy of light incident on the sample. The coefficient *n* depends on the type of interband transitions. For tungsten oxide, *n* = ½, since it is an indirect band gap semiconductor. The experimental optical band gap E*_g_* was determined to be 3.06 eV (Figure 4) and is consistent with data in the literature for tungsten oxide [71,72].

The sensor response to isopropanol, ethanol, and acetone at increasing concentrations (1000–3000 ppm) is presented in the resistance–time plot (Figure 5) and response–concentration curve (Figure 6).

For isopropanol and ethanol, the sensor resistance decreases significantly with increasing gas concentration, indicating strong adsorption of gas molecules onto the sensor surface. For acetone, the resistance shows only a slight reduction, suggesting a weaker interaction between acetone molecules and the sensor material. At higher concentrations, the resistance decreases most prominently for ethanol, followed by isopropanol, while acetone induces only minimal changes.

The sensor response increases linearly with the concentration of ethanol and isopropanol, highlighting their strong interaction with the sensor. At 3000 ppm, the sensor response reaches approximately 2.0 for ethanol and 1.8 for isopropanol. In contrast, the sensor response to acetone remains significantly lower, with minimal increases beyond 1.2 even at 3000 ppm. This indicates limited sensitivity to acetone compared to the other two gases.

The sensor demonstrates high sensitivity to ethanol and isopropanol across the tested concentration range. The linear increase in response with concentration suggests consistent adsorption behavior and strong interaction with the sensor surface. The weaker response to acetone indicates lower adsorption efficiency, possibly due to differences in molecular size, polarity, or binding affinity with the sensor material.

A comparison of the concentration dependence of the WO_3−x_/WS_2_ nanocomposite’s response with the Freundlich isotherm is shown in Figure 6. It has been found that in the studied concentration range, the response can be described by the equation$$S=A\cdot {\left[C\right]}^{1/B},$$
where C is the target gas concentration, and A and B are fitting parameters [73]. The values of A and B are also presented in Figure 6. The calculated detection limits are 500 ppm for ethanol, 600 ppm for isopropanol, and 750 ppm for acetone.

The response and recovery times of the sensor were analyzed for 1000 ppm ethanol and 1000 ppm isopropanol, as shown in Figure 7. The sensor’s behavior for each gas reveals distinct differences in response time and recovery time. The sensor exhibits a gradual increase in response when ethanol is introduced, reaching a maximum value of around 1.35 R_a_/R_g_. Upon the removal of ethanol, the recovery phase follows a nearly symmetrical behavior, with resistance returning to baseline within approximately 20 s (Figure 7a). The sensor shows a faster response to isopropanol compared to ethanol, reaching a peak response of approximately 1.25 R_a_/R_g_. Similarly, the recovery phase is rapid, with the resistance returning to baseline in around 12 s (Figure 7b).

The sensor demonstrates a faster response and recovery for isopropanol compared to ethanol. Specifically, the response and recovery times for isopropanol are shorter by approximately 7–8 s compared to ethanol. This difference may be attributed to the molecular properties of isopropanol, such as its higher vapor pressure and smaller molecular interaction time with the sensor surface, facilitating faster adsorption and desorption dynamics.

The sensor’s resistance response to 2000 ppm isopropanol was evaluated over 10 repeated cycles, as shown in Appendix A. The sensor exhibits a consistent decrease in resistance during each cycle of isopropanol exposure, followed by recovery when the gas is removed. The resistance drops from approximately 65 GΩ to a minimum of 50 GΩ during each exposure phase, indicating a stable and reproducible response to 2000 ppm of isopropanol. The resistance partially recovers to its initial baseline level (~65 GΩ) during the gas removal phase, demonstrating good reversibility. Although slight variations in recovery can be observed across cycles, the overall trend remains stable, indicating the sensor’s reliability under repeated exposure. For recovery, a longer purge of the sample with dried air is necessary. The amplitude of the resistance change remains nearly constant over the 10 cycles, confirming the sensor’s ability to maintain sensitivity and stability over prolonged operation.

Over time, a change in the sensor response to isopropanol was observed. Measurements conducted 8 months before the current ones showed a response of 1.75 to 800 ppm of isopropyl alcohol and 1.2 to ethanol vapors. The results are shown in Appendix A. Current measurements showed that the response to isopropanol of the same concentration is ~1.3 (Figure 6). The response to ethanol (800 ppm) remained almost unchanged (~1.2, as can be seen in Figure 6). Thus, the degradation of sensor characteristics during the detection of isopropanol by 26% over 8 months was demonstrated. These results can be explained by different mechanisms of interaction of the sensor layer with alcohols of different composition.

The responses of the nanocomposite to ammonia in a concentration range of 800–2400 ppm are presented in Appendix A. The response to 1600 ppm of NH_3_ was 1.1, which is significantly lower than the corresponding values for volatile organic compounds. Additionally, incomplete recovery was observed within 2 min, due to the slow desorption of reaction products at room temperature.

Water molecules also play an important role in the processes leading to a sensor response at room temperature [74,75]. Change in sensor baseline resistance when exposed to humidity is shown in Appendix A. The resistance of the sensor significantly decreases upon the introduction of humidity, as shown in the plot. Initially, the resistance remains high (~50 GΩ) under dry conditions. Upon the injection of humidity at around 100 s, a sharp decrease in resistance is observed, reaching a minimum value of approximately 10 GΩ. This decrease is attributed to the interaction of water molecules with the sensor surface, where the water molecules provide free electrons, leading to an increase in conductivity and a corresponding reduction in resistance.

When the humidity is removed (at approximately 200 s), the resistance begins to recover but does not return to its original baseline value. This is due to residual water molecules that remain adsorbed on the sensor surface, continuing to influence its electrical properties.

To eliminate the residual water molecules, a dry gas flow was introduced for 10 min during the experiment. This facilitated the desorption of the water molecules from the sensor surface, enabling the resistance to recover to its original baseline level (~50 GΩ). This process highlights the sensor’s ability to regain its initial state under controlled conditions, ensuring its stability and reusability.

The experimental results demonstrate that humidity strongly affects the resistance of the sensor by reducing it through the adsorption of water molecules. The incomplete recovery without a dry gas flow underscores the need for a desorption step to fully restore the baseline resistance.

The resistance of the sensor shows a significant dependence on relative humidity (RH), as illustrated in Figure 8. When exposed to increasing RH levels (20%, 30%, 40%, 50%, and 60%), the sensor resistance exhibits a clear downward trend.

At 20% RH, the resistance begins to decrease slightly, indicating the initial interaction of water molecules with the sensor surface.

As the RH increases to 30%, 40%, 50%, and 60%, the resistance decreases progressively, reaching values below 50 MΩ at 60% RH.

This behavior suggests that higher humidity levels introduce a greater number of water molecules onto the sensor surface, facilitating more pronounced electron transfer and enhancing conductivity. The response at each RH level is rapid, reflecting the sensor’s sensitivity to moisture changes.

The sensor response to isopropanol at various concentrations (1000–3000 ppm) under increasing RH conditions (0%, 10%, 20%, 30%, and 40%) is presented in the resistance–time (Appendix A) and response–RH plots (Figure 9).

The sensor response decreases as the relative humidity increases, irrespective of isopropanol concentration. For example, at 1000 ppm, the response decreases from ~1.35 at 0% RH to ~1.1 at 40% RH. At higher isopropanol concentrations (e.g., 3000 ppm), the response remains higher but follows the same trend, decreasing from ~1.7 at 0% RH to ~1.25 at 40% RH.

The resistance decreases progressively with increasing RH, as shown in the resistance–time plot. This reduction is due to the adsorption of water molecules, which act as charge carriers and increase the conductivity of the sensor material, leading to lower baseline resistance. The presence of humidity reduces the relative change in resistance caused by isopropanol exposure, thereby lowering the overall response.

The observed reduction in sensor response with increasing RH can be attributed to competitive adsorption between water molecules and isopropanol on the sensor surface. Water molecules occupy active adsorption sites, limiting the adsorption of isopropanol molecules, which weakens the sensor response. Additionally, the presence of humidity introduces additional charge carriers, leading to changes in baseline resistance and reduced signal contrast.

It is known that WO_3_ and WS_2_ are n-type semiconductors, whose resistance decreases when they interact with reducing gases such as alcohols [76]. The operation mechanism of WO_3−x_/WS_2_ nanocomposite is illustrated using its reaction with isopropanol. This reaction consists of two stages. The first one is the adsorption of oxygen molecules on the surface of WO_3−x_/WS_2_ nanostructures and the formation of hydroxyl groups. These reactions can be described by the following equations:(3)$$O_{2\;(gas)}↔O_{2\;(ads)}$$(4)$$O_{2\;(ads)}+e^{-}↔O_{2\;(ads)}^{-}$$

Electrons transfer from the conduction band to the surface energy levels formed by oxygen adsorbed on the surface. At the same time, the bending of the energy bands increases, and the resistance of the semiconductor increases.

The second stage of the sensor operation involves the adsorption of isopropanol molecules on the surface of WO_3−x_/WS_2_ nanostructures, followed by redox reactions between the gas and oxygen molecules, which can be described by the reaction:(5)$$2C_{3}H_{7}OH+9O_{2}^{-}↔{6CO}_{2}+{8H}_{2}O+{9e}^{-}$$

The electrons return into the conduction band, reducing the depletion layer in WO_3−x_/WS_2_ nanostructures. This decreases the resistance of the sensor layer. The reaction products, carbon dioxide and water, are desorbed from the sensor surface.

A comparison of the characteristics of the sensor based on the WO_3−x_/WS_2_ nanocomposite obtained in this study with data from the literature is presented in Table 1. Analysis of the results shows that the sensor layers produced in this study have the advantages of a low operating temperature, fast response, and recovery.

## 3. Experimental Section

WO_3−x_/WS_2_ composite materials were prepared using the hydrothermal method, following the procedure described in reference [81]. Sodium tungstate (Na_2_WO_3_·2H_2_O) was dissolved in distilled water using a magnetic stirrer to create a solution with a concentration of 30 mM. To adjust the acidity of the solution, hydrochloric acid (HCl) (2 M) was added, resulting in a pH of 3.75. A total of 900 mg of L-cysteine was then added to the solution and mixed until the mixture became transparent and all components were dissolved. The resulting solution was then placed in an autoclave and heated to 200 °C for 24 h. After cooling to room temperature, the blue precipitate was separated using centrifugation and washed three times with distilled water. It was then dried in a muffle oven at 75 °C.

Glass ceramic (ST-50-1-1-0.6, roughness—0.032 μm) substrates with aluminum electrodes were used for applying WO_3−x_/WS_2_ sensor layers (Figure 10). To deposit the WO_3−x_/WS_2_ sensor layer on a substrate, approximately 0.2 g of powder was mixed with 2 mL of isopropyl alcohol and placed in an ultrasonic bath for 30 min. After that, 20 μL of the resulting solution was applied on pre-cleaned substrates by spin-coating. To form a thin layer, spin-coating was carried out at 2000 rpm for 15 s, and then at 3000 rpm for the same time. This process was repeated 20 times until a relatively homogeneous, dense layer covered the entire area of the substrate. The resulting layer was then dried for 30 min at 75 °C. The synthesis process is shown in Figure 11.

The thickness of the sensor layer deposited on the substrate was measured using an optical microscope, POLAM P-312 (LOMO, Saint Petersburg, Russia). The surface morphology of the WO_3−x_/WS_2_ nanocomposites was examined by scanning electron microscopy using a Zeiss Supra 25 scanning electron microscope (Zeiss, Oberkochen, Germany). EDX analysis was used to determine the elemental composition of the sample surface.

Diagnostics by X-ray photoelectron spectroscopy was performed using a K-Alpha spectrometer (Thermo Scientific, Waltham, MA, USA), equipped with a monochromatic Al Ka X-ray source (λ = 1486.6 eV). The residual pressure in the analytical chamber was approximately 4 × 10^−9^ mbar. The XPS measurements were conducted with a resolution of 1 eV for the survey spectrum, and the core level spectra of oxygen, sulfur, and tungsten were obtained with a resolution of 0.1 eV. In order to evaluate the contribution of different chemical environments of elements, the obtained spectra were deconvolved using Gaussian functions. A linear background was preliminarily subtracted.

To determine the optical band gap width of the WO_3−x_/WS_2_ nanocomposite, the powder was deposited on a quartz glass substrate using spin-coating. Optical absorbance spectra were then obtained using a PE-5400UF spectrophotometer (EKROSCHIM, Saint Petersburg, Russia).

The system is designed to generate vapor flows and measure the gas-sensing properties of materials by monitoring changes in electrical resistance under exposure to target vapors or humidity. The vapor generation setup includes an air pump that supplies dried air into three separate branches. In the first branch, the airflow is regulated by MFC 1, which works together with MFC 2 and MFC 3 to control the concentration of gas vapors and relative humidity before the flow enters a mixing chamber. The second branch uses MFC 2 to direct air through a bubbler containing the target solution, such as isopropanol, ethanol, or acetone, where the airflow pressure causes the liquid to transition into vapor. Similarly, the third branch generates water vapor by passing air through a bubbler filled with distilled water. These vapor flows are uniformly mixed in a mixing chamber before being introduced into a sample chamber for testing. To calculate the pressure of saturated vapors of the liquid in the bubbler, the Antoine equation is used.

The measurement system consists of the sample chamber (volume 0.193 L), a power supply, control components, and measuring instruments (Figure 12). An Arduino controller manages system parameters, including valve operations and environmental settings, ensuring flexible and precise control. The electrical resistance of the material is measured using a Picoammeter 6485 (Keithley, Cleveland, OH, USA) for small current detection. Measurement data are transferred to a computer for further analysis and processing. The sample is purged with a stream of dried air for 10 min before taking measurements. All measurements are conducted at an ambient temperature of 22–25 °C.

The response is calculated as the ratio of the resistance in the sample in air to the resistance when the sample is exposed to the target gas (R_air_/R_gas_). The response and recovery times are calculated as the time it takes for the resistance to change by 90% of the total amplitude of the change [82,83]. The detection limit is estimated as 3[C]/((R_air_ − R_gas_)/σ), where C is the concentration of the target gas, and σ is the signal fluctuation [84].

## 4. Conclusions

This work is dedicated to the development of sensor layers based on WO_3−x_/WS_2_ nanocomposites capable of detecting gases at room temperature. WO_3−x_/WS_2_ nanostructures consisting of one-dimensional (1D) and two-dimensional (2D) nano-objects were synthesized by the hydrothermal method. The thickness of the sensor layer deposited on the substrate was approximately 22 to 23 μm. The optical band gap of the sample was found to be 3.06 eV. XPS and EDX studies characterizing the elemental composition of the material suggested that tungsten, sulfur, and oxygen were present in the sample. It was shown that the sample mainly contained WO_3−x_, and traces of WS_2_ were also detected. The WO_3−x_/WS_2_ nanostructures showed a gas-sensing response to isopropanol and ethanol at room temperature. Therefore, the sensor layers do not require elevated operating temperatures. This reduces the cost and simplifies the production of gas analytical devices.

## Figures and Tables

**Figure 1 molecules-30-00566-f001:**
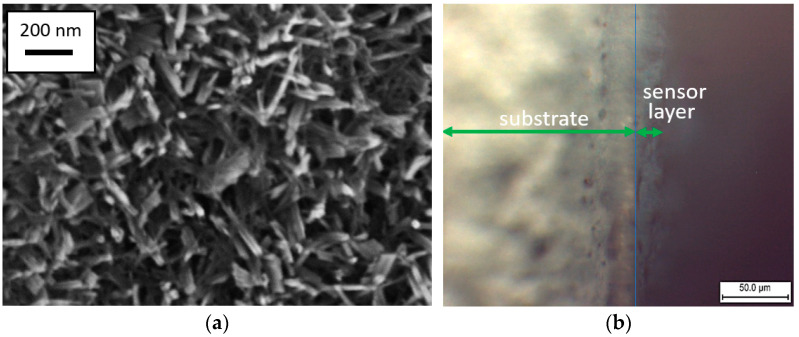
(**a**) SEM image of WO_3−x_/WS_2_ nanocomposite, (**b**) cleavage of substrate with WO_3−x_/WS_2_ sensor layer.

**Figure 2 molecules-30-00566-f002:**
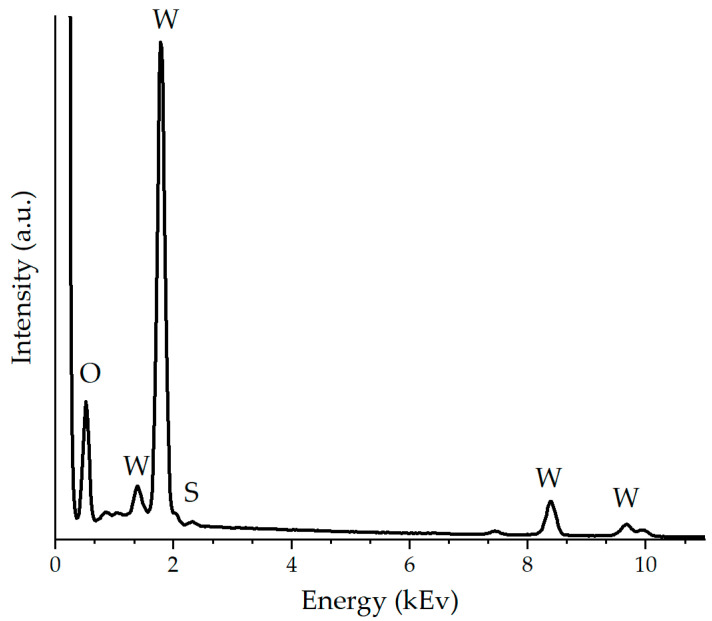
EDX spectrum of WO_3−x_/WS_2_ nanocomposite.

**Figure 3 molecules-30-00566-f003:**
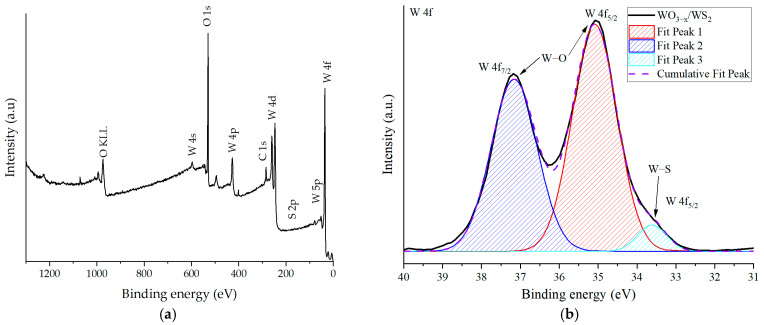
XPS spectra WO_3−x_/WS_2_ nanostructure: survey (**a**), W 4f (**b**), O 1s (**c**), S 2p (**d**) levels.

**Figure 4 molecules-30-00566-f004:**
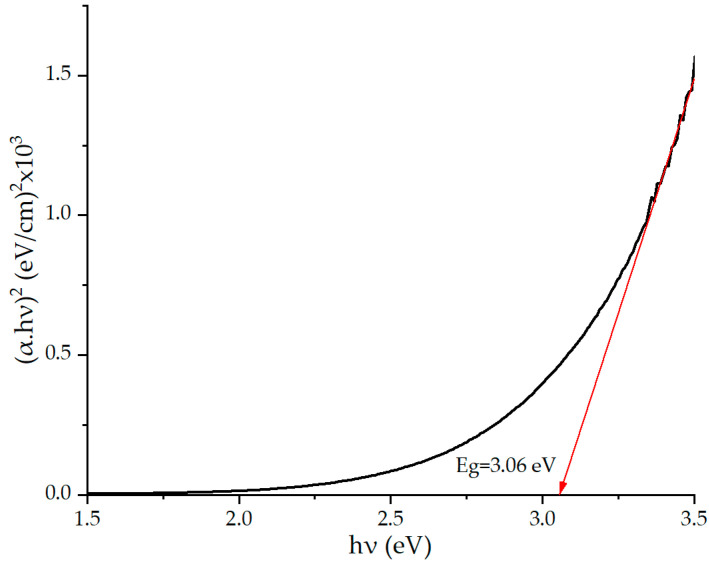
Tauc plot of WO_3−x_/WS_2_ nanocomposite.

**Figure 5 molecules-30-00566-f005:**
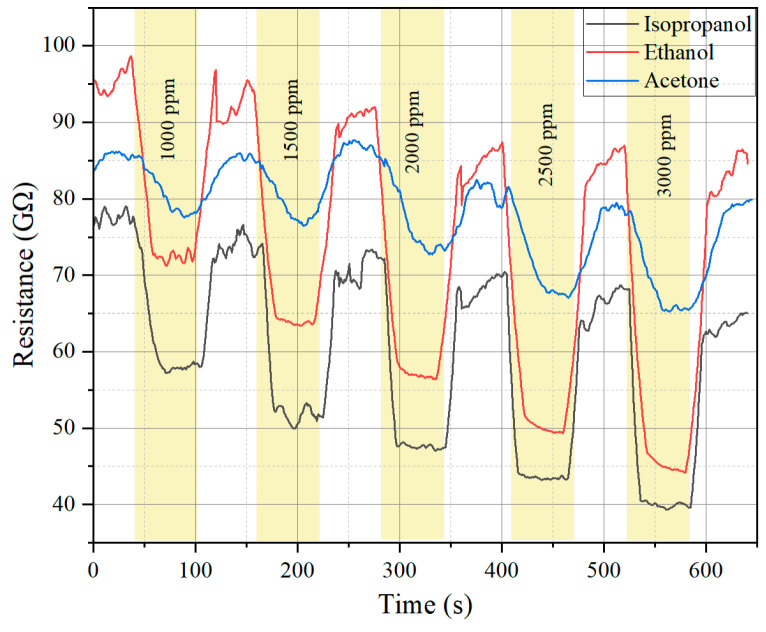
Sensor resistance when exposed to different target gases.

**Figure 6 molecules-30-00566-f006:**
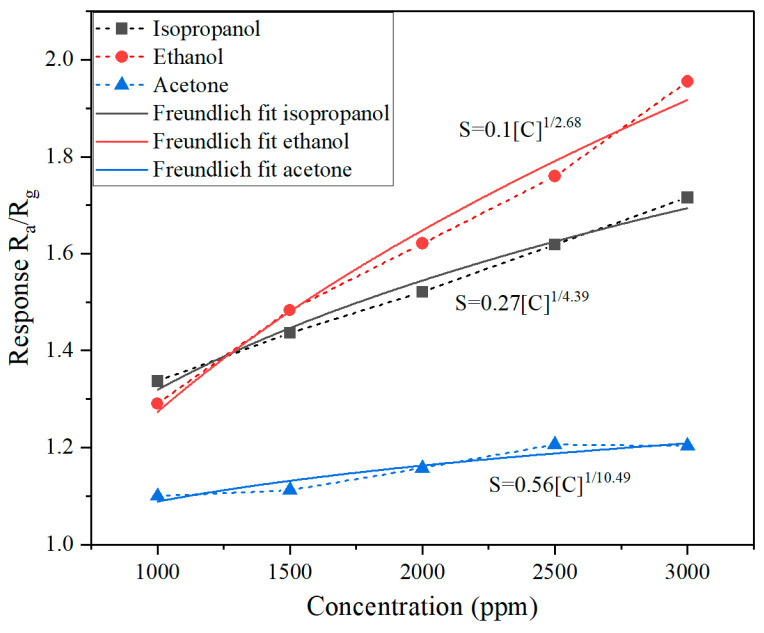
Sensor response when exposed to different target gas concentrations.

**Figure 7 molecules-30-00566-f007:**
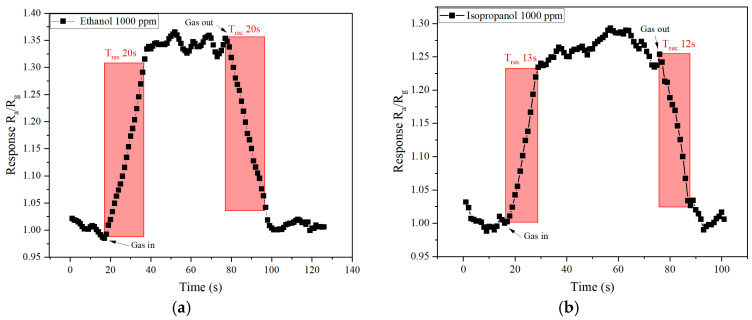
Response and recovery times of WO_3−x_/WS_2_ nanocomposite when detecting (**a**) ethanol and (**b**) isopropanol.

**Figure 8 molecules-30-00566-f008:**
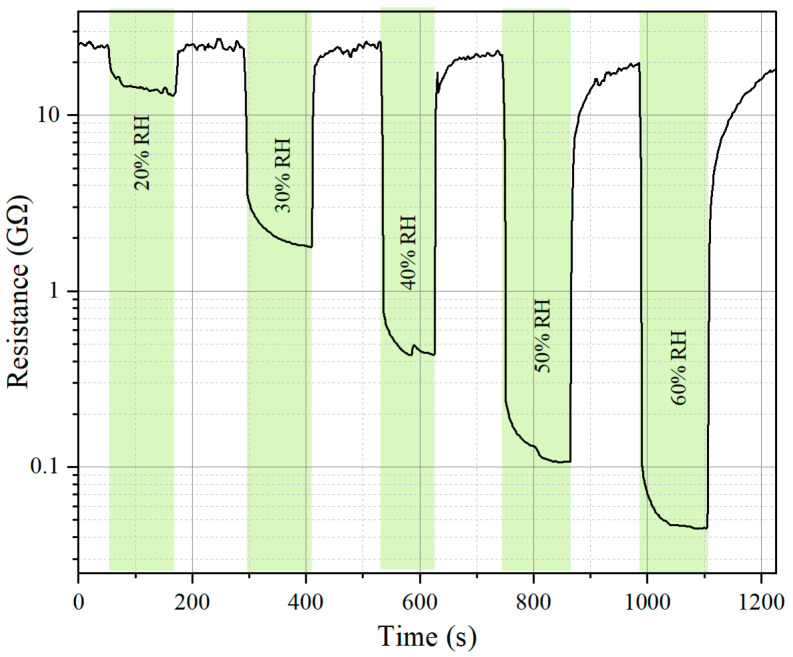
Influence of humidity on sensor resistance.

**Figure 9 molecules-30-00566-f009:**
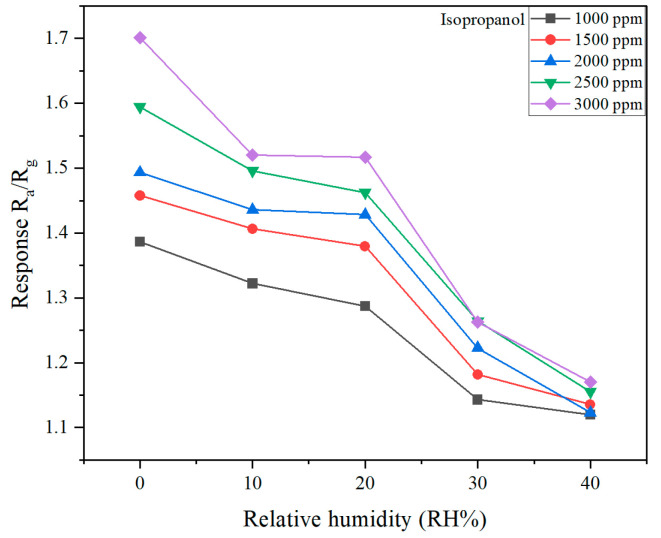
Influence of relative humidity on the sensor response.

**Figure 10 molecules-30-00566-f010:**
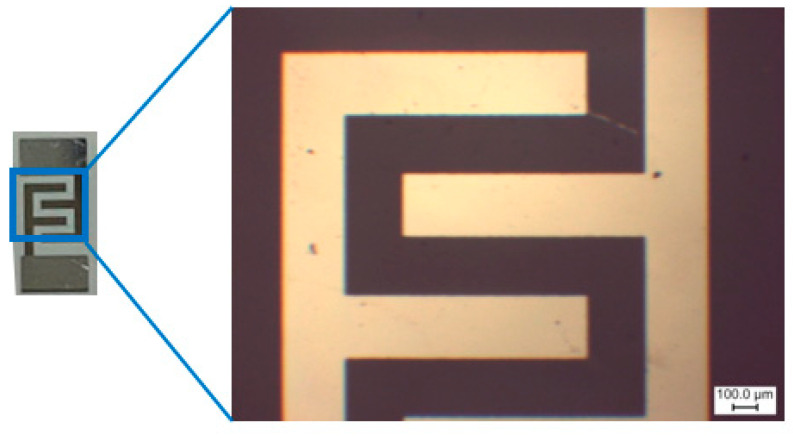
The substrate used for depositing the sensor layer.

**Figure 11 molecules-30-00566-f011:**
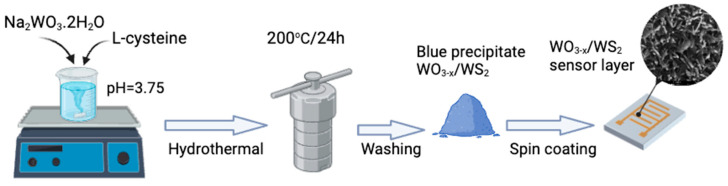
Stages of WO_3−x_/WS_2_ nanocomposite synthesis.

**Figure 12 molecules-30-00566-f012:**
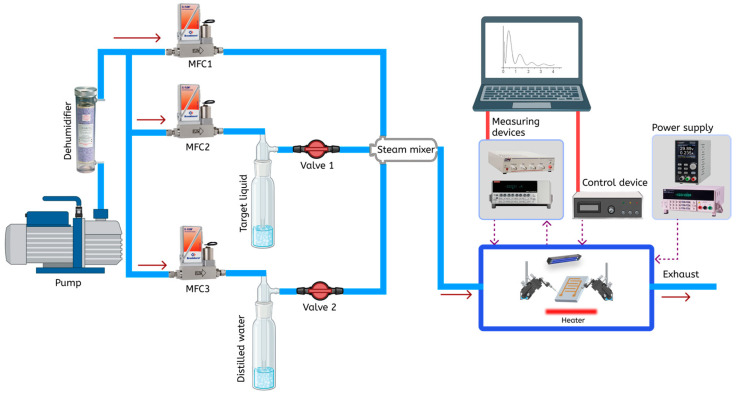
System for generating vapor flows and measuring sensor gas properties.

**Table 1 molecules-30-00566-t001:** WO_3_ gas sensors for ethanol and propanol detection.

Sensing Material	Target Gas/Concentration	Response, R_a_/R_g_	Working Temperature, °C	Response Time, s	Recovery Time, s	Reference
WO_3_ nanoflakes	Ethanol/5ppm	7.5	250	9 min 40 s	21 min	[77]
WO_3_ nanoplates	Ethanol/10 ppm	8	300	4 s	10 s	[61]
Nanostructured WO_3_ thin films	Ethanol/100 ppm	8	400	12 s	56 s	[78]
Hierarchical flower-like co-doped WO_3_ nanoplates	Ethanol/100 ppm	2.3	250	-	-	[79]
WO_3_ nanorods	Ethanol/100 ppm	26.48	160	1 s	30 s	[62]
Porous WO_3_ nanolamellae	Ethanol/10 ppm	11	200	8.5 s	6.5 s	[63]
SnO_2_-decorated WO_3_ structures	Propanol/50 ppm	242%	275	150 s	150 s	[80]
WO_3−x_/WS_2_ nanocomposites	Ethanol/1000 ppm	1.35	Room temperature	20 s	20 s	This work
	Isopropanol/1000 ppm	1.25		13 s	12 s	

## Data Availability

Data are contained within the article.

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
