# Peer review of "WO3−x/WS2 Nanocomposites for Fast-Response Room Temperature Gas Sensing"

_molecules, 2025, doi:10.3390/molecules30030566_

Round 1
Reviewer 1 Report
Comments and Suggestions for Authors
The authors present in their manuscript a room-temperature study regarding detection of alcohol vapors using WO3-x/WS2 nanocomposites.
Prior to manuscript acceptance the authors must provide answers to the following issues:
1. The study is well written and the experimental section is thorough, BUT as it is common in the sensing domain, you must cover the so called 3S parameters of a sensing device: sensibility, selectivity and stability. Your study covers only two out of three, namely sensibility/selectivity towards target gases. You must present a separate figure with sensor (response) stability during a reasonable period of time (at least 3-6 months between measurements).
2. Giving the fact that you perform the measurements at Tw= room temperature you must provide information regarding humidity influence over sensor response, as your adsorbed water molecules are important contaminants in the case of your measurements parameters. Was your surface at least "cleaned" with dry carrier gas prior to sensing cycles? Was your sensor heated up to remove that adsorbed humidity at the beginning of your experiments? Please provide those details in the manuscript.
3. Please provide a separate figure with sensor response to relative humidity at Tw= room temperature.
4. What kind of glass substrate did you used in your sensor fabrication step? Was it commercially available? Was your glass substrate porous? Why not use commercially available porous alumina substrate instead? Please provide more details in the experimental section, as this is a KEY feature for your sensing device.
Author Response
We would like to thank the reviewers for their careful and detailed reports. We appreciate their comments and suggestions and have made every effort to address them in the revised manuscript. The changes made in the revised version are clearly indicated.
Comments 1. The study is well written and the experimental section is thorough, BUT as it is common in the sensing domain, you must cover the so called 3S parameters of a sensing device: sensibility, selectivity and stability. Your study covers only two out of three, namely sensibility/selectivity towards target gases. You must present a separate figure with sensor (response) stability during a reasonable period of time (at least 3-6 months between measurements).
Response 1. Thank you for your comment. Over time, a change in the sensor response to isopropanol has been observed. Measurements conducted 8 months before the current ones showed a response to 800 ppm of isopropyl alcohol of 1.75, and to ethanol vapors of 1.2. The results are shown in Figures S2 and S3. Repeated measurements 8 months later showed that the response to isopropanol of the same concentration is ~ 1.3. The response to ethanol (800 ppm) remained almost unchanged (~ 1.2). Thus, the degradation of sensor characteristics during the detection of isopropanol by 26% over 8 months was demonstrated. These results can be explained by different mechanisms of interaction of the sensor layer with alcohols of different composition.
Relevant additions have been made to manuscript.
Comments 2. Giving the fact that you perform the measurements at Tw= room temperature you must provide information regarding humidity influence over sensor response, as your adsorbed water molecules are important contaminants in the case of your measurements parameters. Was your surface at least "cleaned" with dry carrier gas prior to sensing cycles? Was your sensor heated up to remove that adsorbed humidity at the beginning of your experiments? Please provide those details in the manuscript.
Response 2. Thank you for your comment. The sample was purged with a stream of dried air for 10 minutes before taking measurements. The sample was heated only to a temperature of 70 ° C, since WS2 begins to oxidize at higher temperatures.
Change in sensor baseline resistance when exposed to humidity is shown in Fig. S1. The resistance of the sensor significantly decreases upon the introduction of humidity, as shown in the plot. Initially, the resistance remains high (~50 GΩ) under dry conditions. Upon the injection of humidity at around 100 seconds, a sharp decrease in resistance is observed, reaching a minimum value of approximately 10 GΩ. This decrease is attributed to the interaction of water molecules with the sensor surface, where the water molecules provide free electrons, leading to an increase in conductivity and a corresponding reduction in resistance.
When the humidity is removed (at approximately 200 seconds), the resistance begins to recover but does not return to its original baseline value. This is due to residual water molecules that remain adsorbed on the sensor surface, continuing to influence its electrical properties.
To eliminate the residual water molecules, a dry gas flow was introduced for 10 minutes during the experiment. This facilitated the desorption of the water molecules from the sensor surface, enabling the resistance to recover to its original baseline level (~50 GΩ). This process highlights the sensor’s ability to regain its initial state under controlled conditions, ensuring its stability and reusability.
The experimental results demonstrate that humidity strongly affects the resistance of the sensor by reducing it through the adsorption of water molecules. The incomplete recovery without a dry gas flow underscores the need for a desorption step to fully restore the baseline resistance.
Relevant additions have been made to manuscript.
Comments 3. Please provide a separate figure with sensor response to relative humidity at Tw= room temperature.
Response 3. Thank you for your comment. The resistance of the sensor shows a significant dependence on relative humidity (RH), as illustrated in the figure 8. When exposed to increasing RH levels (20%, 30%, 40%, 50%, and 60%), the sensor resistance exhibits a clear downward trend.
At 20% RH, the resistance begins to decrease slightly, indicating the initial interaction of water molecules with the sensor surface.
As the RH increases to 30%, 40%, 50%, and 60%, the resistance decreases progressively, reaching values below 50 MΩ at 60% RH.
This behavior suggests that higher humidity levels introduce a greater number of water molecules onto the sensor surface, facilitating more pronounced electron transfer and enhancing conductivity. The response at each RH level is rapid, reflecting the sensor's sensitivity to moisture changes.
Relevant additions have been made to manuscript.
Comments 4. What kind of glass substrate did you used in your sensor fabrication step? Was it commercially available? Was your glass substrate porous? Why not use commercially available porous alumina substrate instead? Please provide more details in the experimental section, as this is a KEY feature for your sensing device.
Response 4. Thank you for your comment. Commercial glass ceramic substrates (ST-50-1-1-0,6) were used in experiments. The roughness was 0,032 μm, the substrate surface was smooth.
Relevant additions have been made to experimental section.
Reviewer 2 Report
Comments and Suggestions for Authors
The work presented by Svetlana S. Nalimova et al, deals with the development of a WO3-based chemiresistors for ethanol and isopropanol detection. To reduce the working temperature and to improve the sensing performances they proposed to create an heterojunction between WO3 and WS2. After the sample preparation, they perform characterization with SEM, EDX and XPS and then they proceeded with gas exposures.
The work could be of interest, but I do not think that it can be published in the current form. The characterization is quite superficial, and the sensing performances must be investigated more in depth. Only 2 exposures for each gas (ethanol and 2-propanol) are presented and the tested concentrations are quite high (600 ppm and 800 ppm). Calibration curves, limit of detection, selectivity, reproducibility, stability over time, benchmarking with literature and influence of humidity are completely missing and must be investigated and discussed, before the present work can be considered for publication. Additionally, the proposed sensing mechanism is not supported by experimental evidence.
I am afraid but considering all the missing investigation, at this time I suggest rejection.
In details, some comments that can help the authors to improve the manuscript:
1) The introduction is not well focused. It is not clear how the proposed work can improve the state of art in the WO3-based sensors development.
2) Consider merging figure 3 and 4.
3) Figure 4c and main text: the S 2p peak should be taken with higher accumulations. Indeed, the signal to noise ratio is quite high, and the contribution of the 2p3/2 is completely in the noise; there is no difference in the intensity of the feature at 166.5 eV, which is considered as noise, and the one at 160.5 eV, which is assigned to the S 2p3/2 contribution. Please, present a spectrum with a lower signal-to-noise ratio.
4) Figures 6 and 7 and text: baseline of the signal before the gas exposure should be presented. Additionally, the fluctuation of the signal during the exposure should be justified and discussed. Finally, how did the authors evaluate response and recovery time?
5) A sequence of dynamical curves (response vs time) for different concentrations should be presented, to prove the capability of the sensors to work in a reliable manner.
6) Calibration curves should be always drawn; additionally, the tested concentrations are quite high for many applications of chemiresistors. Therefore, it is suggested to perform additional exposures to lower concentrations, to draw the calibration curve. Then calibration curves should be compared with adsorption isotherms (i.e. Freundlich or Langmuir…etc…) and the results should be properly discussed. Moreover, detection limit should be evaluated (see for instance the formula reported in: Freddi, S., Marzuoli, C., Pagliara, S., Drera, G., & Sangaletti, L. (2023). Targeting biomarkers in the gas phase through a chemoresistive electronic nose based on graphene functionalized with metal phthalocyanines. RSC advances, 13(1), 251-263).
7) Selectivity must be proven at least for the most common interfering gases, such as ammonia, nitrogen dioxide, carbon dioxide, acetone.
8) Benchmarking with literature for response/recovery time, response, and detection limit should be performed (see for instance: Spagnoli, E., Krik, S., Fabbri, B., Valt, M., Ardit, M., Gaiardo, A., ... & Guidi, V. (2021). Development and characterization of WO3 nanoflakes for selective ethanol sensing. Sensors and Actuators B: Chemical, 347, 130593. Chen, D., Hou, X., Wen, H., Wang, Y., Wang, H., Li, X., ... & Gao, L. (2009). The enhanced alcohol-sensing response of ultrathin WO3 nanoplates. Nanotechnology, 21(3), 035501. Ahmad, M. Z., Sadek, A. Z., Ou, J. Z., Yaacob, M. H., Latham, K., & Wlodarski, W. (2013). Facile synthesis of nanostructured WO3 thin films and their characterization for ethanol sensing. Materials Chemistry and Physics, 141(2-3), 912-919. Acharyya, S., Manna, B., Nag, S., & Guha, P. K. (2019, October). WO 3 nanoplates based chemiresistive sensor device for selective detection of 2-propanol. In 2019 IEEE SENSORS (pp. 1-4). IEEE.)
9) Reproducibility of the response should be assessed, performing cycles of exposures at the same concentration. Additionally, also stability over time needs to be considered.
10) Experimental evidence to support the proposed sensing mechanism should be reported, exposing the sensors to different humidity, with and without the target gas. Moreover, it is quite uncommon to define the p-type/n-type nature of a gas sensor considering exposures to alcohols. Usually, it is done considering strong reducing or oxidizing gases, such as NH3 or NO2. Authors should perform exposures to at least one of the two suggested gases, to completely prove the p-type nature of the sensor. Finally, authors can consider also the following works to discuss the sensing mechanism: Patrick, D. S., Bharathi, P., Kamalakannan, S., Archana, J., Navaneethan, M., & Mohan, M. K. (2024). Confined oxidation of 2D WS2 nanosheets forming WO3/WS2 nanocomposites for room temperature NO2 gas sensing application. Applied Surface Science, 642, 158554. Qin, F., Gao, J., Jiang, L., Fan, J., Sun, B., Fan, Y., ... & Shi, K. (2023). Biomorphic WO3@ WS2 heterojunction composites for enhanced NO2 gas-sensing performance at room temperature. Applied Surface Science, 615, 156338.
11) Crystalline phase of the WO3 can influence the sensing performances, therefore it is recommended to perform XRD measurements, to disclose the crystalline phase of the developed sensor.
12) Experimental section: Info on the XPS analysis should be added: which is the overall resolution of the acquired spectra? Additionally, which fitting function has been used (Voigt?) and which background feature?
13) Gas measurements: info on the chamber volume, on the relative humidity and temperature in the chamber during the exposures, exposure time and recovery time and how the chamber is purged after the exposures should be added. How is the concentration evaluated? Finally, response is defined as (Iair/Igas) at line 214, but in the rest of the manuscript response is defined considering the resistance and not the current.
Author Response
We would like to thank the reviewers for their careful and detailed reports. We appreciate their comments and suggestions and have made every effort to address them in the revised manuscript. The changes made in the revised version are clearly indicated.
Comments 1. The introduction is not well focused. It is not clear how the proposed work can improve the state of art in the WO3-based sensors development.
Response 1. Thank you for your comment. The following text has been added to the introduction.
Thus, in most studies, the response was achieved at elevated temperatures, except in cases where NO2 was used as the detected gas. The development of composite sensor layers based on WO3, showing a response at room temperature, is an urgent task in modern gas sensors.
Comments 2. Consider merging figure 3 and 4.
Response 2. Thank you for your comment. The figures are merged.
Comments 3. Figure 4c and main text: the S 2p peak should be taken with higher accumulations. Indeed, the signal to noise ratio is quite high, and the contribution of the 2p3/2 is completely in the noise; there is no difference in the intensity of the feature at 166.5 eV, which is considered as noise, and the one at 160.5 eV, which is assigned to the S 2p3/2 contribution. Please, present a spectrum with a lower signal-to-noise ratio.
Response 3. Thank you for your comment. The deconvolution of S2p spectrum was carried out taking into account the analysis of the spectra of the other elements. Thus, the peak at a binding energy of ~159 eV is not interpreted by the contribution of sulfur, so it was not taken into account. The peak with an energy of 166.5 eV may be associated with the contribution of sulfur bound to oxygen [http://dx.doi.org/10.17576/jsm-2018-4708-33], but there is no corresponding contribution on the spectrum of the core oxygen level, so it was also not taken into account.
Comments 4. Figures 6 and 7 and text: baseline of the signal before the gas exposure should be presented. Additionally, the fluctuation of the signal during the exposure should be justified and discussed. Finally, how did the authors evaluate response and recovery time?
Response 4. Thank you for your comment. The response and recovery times of the sensor were analyzed for 1000 ppm ethanol and 1000 ppm isopropanol, as shown in the figure 7. The sensor's behavior for each gas reveals distinct differences in response time and recovery time:
Ethanol (1000 ppm):
- Response time: ~20 seconds
- Recovery time: ~20 seconds
- The sensor exhibits a gradual increase in response when ethanol is introduced, reaching a maximum value around 1.35 Ra/Rg. Upon removal of ethanol, the recovery phase follows a nearly symmetrical behavior, with resistance returning to baseline within approximately 20 seconds.
Isopropanol (1000 ppm):
- Response time: ~13 seconds
- Recovery time: ~12 seconds
- The sensor shows a faster response to isopropanol compared to ethanol, reaching a peak response of approximately 1.25 Ra/Rg. Similarly, the recovery phase is rapid, with the resistance returning to baseline in around 12 seconds.
The sensor demonstrates a faster response and recovery for isopropanol compared to ethanol. Specifically, the response and recovery times for isopropanol are shorter by approximately 7–8 seconds compared to ethanol. This difference may be attributed to the molecular properties of isopropanol, such as its higher vapor pressure and smaller molecular interaction time with the sensor surface, facilitating faster adsorption and desorption dynamics.
The response and recovery times were calculated as the time it took for the resistance to change by 90% of the total amplitude of the change [77,78]. The corresponding description has been added to the last paragraph of experimental section.
Relevant additions have been made to manuscript.
Comments 5. A sequence of dynamical curves (response vs time) for different concentrations should be presented, to prove the capability of the sensors to work in a reliable manner.
Response 5. Thank you for your comment. The sensor response to isopropanol, ethanol, and acetone at increasing concentrations (1000–3000 ppm) is presented in the resistance-time plot (Figure 5) and response-concentration curve (Figure 6).
For isopropanol and ethanol, the sensor resistance decreases significantly with increasing gas concentration, indicating strong adsorption of gas molecules onto the sensor surface. For acetone, the resistance shows only a slight reduction, suggesting a weaker interaction between acetone molecules and the sensor material. At higher concentrations, the resistance decreases most prominently for ethanol, followed by isopropanol, while acetone induces only minimal changes.
The sensor response increases linearly with the concentration of ethanol and isopropanol, highlighting their strong interaction with the sensor. At 3000 ppm, the sensor response reaches approximately 2.0 for ethanol and 1.8 for isopropanol. In contrast, the sensor response to acetone remains significantly lower, with minimal increase beyond 1.2 even at 3000 ppm. This indicates limited sensitivity to acetone compared to the other two gases.
The sensor demonstrates high sensitivity to ethanol and isopropanol across the tested concentration range. The linear increase in response with concentration suggests consistent adsorption behavior and strong interaction with the sensor surface. The weaker response to acetone indicates lower adsorption efficiency, possibly due to differences in molecular size, polarity, or binding affinity with the sensor material.
Relevant additions have been made to manuscript.
Comments 6. Calibration curves should be always drawn; additionally, the tested concentrations are quite high for many applications of chemiresistors. Therefore, it is suggested to perform additional exposures to lower concentrations, to draw the calibration curve. Then calibration curves should be compared with adsorption isotherms (i.e. Freundlich or Langmuir…etc…) and the results should be properly discussed. Moreover, detection limit should be evaluated (see for instance the formula reported in: Freddi, S., Marzuoli, C., Pagliara, S., Drera, G., & Sangaletti, L. (2023). Targeting biomarkers in the gas phase through a chemoresistive electronic nose based on graphene functionalized with metal phthalocyanines. RSC advances, 13(1), 251-263).
Response 6. Thank you for your comment. Measurements were carried out over a concentration range of 1000 to 3000 ppm for the target gases (Figure 6). The experimental response at lower concentrations would have a large error due to their small values. To evaluate these values, data obtained from fitting can be used.
A comparison of the concentration dependence of the WO3-x/WS2 nanocomposite's response with the Freundlich isotherm is shown in Figure 6. It has been found that in the studied concentration range, the response can be described by the equation S = A · [C]B. The values of A and B, which are the fitting parameters, are also shown in Figure 6.
Relevant additions have been made to manuscript.
Comments 7. Selectivity must be proven at least for the most common interfering gases, such as ammonia, nitrogen dioxide, carbon dioxide, acetone.
Response 7. Thank you for your comment. An acetone response study has been conducted. Due to the limitations of the laboratory setup, which only allows vapors of bubbled liquids to be introduced into the chamber, this study is focused on the response to volatile organic compounds and water.
Comments 8. Benchmarking with literature for response/recovery time, response, and detection limit should be performed (see for instance: Spagnoli, E., Krik, S., Fabbri, B., Valt, M., Ardit, M., Gaiardo, A., ... & Guidi, V. (2021). Development and characterization of WO3 nanoflakes for selective ethanol sensing. Sensors and Actuators B: Chemical, 347, 130593. Chen, D., Hou, X., Wen, H., Wang, Y., Wang, H., Li, X., ... & Gao, L. (2009). The enhanced alcohol-sensing response of ultrathin WO3 nanoplates. Nanotechnology, 21(3), 035501. Ahmad, M. Z., Sadek, A. Z., Ou, J. Z., Yaacob, M. H., Latham, K., & Wlodarski, W. (2013). Facile synthesis of nanostructured WO3 thin films and their characterization for ethanol sensing. Materials Chemistry and Physics, 141(2-3), 912-919. Acharyya, S., Manna, B., Nag, S., & Guha, P. K. (2019, October). WO 3 nanoplates based chemiresistive sensor device for selective detection of 2-propanol. In 2019 IEEE SENSORS (pp. 1-4). IEEE.)
Response 8. Thank you for your comment. A comparison of the characteristics of the sensor based on WO3-x/WS2 nanocomposite, obtained in this study, with literature data is presented in Table 1. Analysis of the results shows that the sensor layers produced in this study have the advantages of a low operating temperature, fast response, and recovery.
Relevant additions have been made to manuscript.
Comments 9. Reproducibility of the response should be assessed, performing cycles of exposures at the same concentration. Additionally, also stability over time needs to be considered.
Response 9. Thank you for your comment. The sensor's resistance response to 2000 ppm isopropanol was evaluated over 10 repeated cycles, as shown in the figure S1.
The sensor exhibits a consistent decrease in resistance during each cycle of isopropanol exposure, followed by recovery when the gas is removed. The resistance drops from approximately 65 GΩ to a minimum of 50 GΩ during each exposure phase, indicating a stable and reproducible response to 2000 ppm IPA.
The resistance partially recovers to its initial baseline level (~65 GΩ) during the gas removal phase, demonstrating good reversibility. Although slight variations in recovery can be observed across cycles, the overall trend remains stable, indicating the sensor's reliability under repeated exposure.
The amplitude of the resistance change remains nearly constant over the 10 cycles, confirming the sensor's ability to maintain sensitivity and stability over prolonged operation.
Over time, a change in the sensor response to isopropanol has been observed. Measurements conducted 8 months before the current ones showed a response to 800 ppm of isopropyl alcohol of 1.75, and to ethanol vapors of 1.2. The results are shown in Figures S2 and S3. Repeated measurements 8 months later showed that the response to isopropanol of the same concentration is ~ 1.3. The response to ethanol (800 ppm) remained almost unchanged (~ 1.2). Thus, the degradation of sensor characteristics during the detection of isopropanol by 26% over 8 months was demonstrated. These results can be explained by different mechanisms of interaction of the sensor layer with alcohols of different composition.
Relevant additions have been made to manuscript.
Comments 10. Experimental evidence to support the proposed sensing mechanism should be reported, exposing the sensors to different humidity, with and without the target gas. Moreover, it is quite uncommon to define the p-type/n-type nature of a gas sensor considering exposures to alcohols. Usually, it is done considering strong reducing or oxidizing gases, such as NH3 or NO2. Authors should perform exposures to at least one of the two suggested gases, to completely prove the p-type nature of the sensor. Finally, authors can consider also the following works to discuss the sensing mechanism: Patrick, D. S., Bharathi, P., Kamalakannan, S., Archana, J., Navaneethan, M., & Mohan, M. K. (2024). Confined oxidation of 2D WS2 nanosheets forming WO3/WS2 nanocomposites for room temperature NO2 gas sensing application. Applied Surface Science, 642, 158554. Qin, F., Gao, J., Jiang, L., Fan, J., Sun, B., Fan, Y., ... & Shi, K. (2023). Biomorphic WO3@ WS2 heterojunction composites for enhanced NO2 gas-sensing performance at room temperature. Applied Surface Science, 615, 156338.
Response 10. Thank you for your comment. The discrepancy noted in comment 12 regarding the experimental description and data has been eliminated. The response has been defined as Ra/Rg. Based on the results, the sample exhibits an n-type response, consistent with the classical model of oxide semiconductor interaction with reducing gases. Therefore, the last paragraph and Figure 8 have been removed from the section 2 (Results and discussion).
The sensor response to isopropanol at various concentrations (1000–3000 ppm) under increasing relative humidity (RH) conditions (0%, 10%, 20%, 30%, and 40%) is presented in the resistance-time (Figure S5) and response-RH plots (Figure 9).
The sensor response decreases as the relative humidity increases, irrespective of isopropanol concentration. For example, at 1000 ppm, the response decreases from ~1.35 at 0% RH to ~1.1 at 40% RH. At higher isopropanol concentrations (e.g., 3000 ppm), the response remains higher but follows the same trend, decreasing from ~1.7 at 0% RH to ~1.25 at 40% RH.
The resistance decreases progressively with increasing RH, as shown in the resistance-time plot. This reduction is due to the adsorption of water molecules, which act as charge carriers and increase the conductivity of the sensor material, leading to lower baseline resistance. The presence of humidity reduces the relative change in resistance caused by isopropanol exposure, thereby lowering the overall response.
The resistance decreases progressively with increasing RH, as shown in the resistance-time plot. This reduction is due to the adsorption of water molecules, which act as charge carriers and increase the conductivity of the sensor material, leading to lower baseline resistance. The presence of humidity reduces the relative change in resistance caused by isopropanol exposure, thereby lowering the overall response.
The observed reduction in sensor response with increasing RH can be attributed to competitive adsorption between water molecules and isopropanol on the sensor surface. Water molecules occupy active adsorption sites, limiting the adsorption of isopropanol molecules, which weakens the sensor response. Additionally, the presence of humidity introduces additional charge carriers, leading to changes in baseline resistance and reduced signal contrast.
Relevant additions have been made to manuscript.
Comments 11. Crystalline phase of the WO3 can influence the sensing performances, therefore it is recommended to perform XRD measurements, to disclose the crystalline phase of the developed sensor.
Response 11. Thank you for your comment. The resulting powder composite material was found to be insufficient for powder diffraction analysis, and the use of XRD for films on a glass ceramic substrate was inefficient for determining the phases. Therefore, due to the fact that the interaction of target gases with the sensor layers occurs on their surface, we chose XPS as the analysis method.
Comments 12. Experimental section: Info on the XPS analysis should be added: which is the overall resolution of the acquired spectra? Additionally, which fitting function has been used (Voigt?) and which background feature?
Response 12. Thank you for your comment. The XPS measurements were conducted with a resolution of 1 eV for the survey spectrum, and the core level spectra of oxygen, sulfur, and tungsten were obtained with a resolution of 0.1 eV. In order to evaluate the contribution of different chemical environments of elements, the obtained spectra were deconvolved using Gaussian functions. A linear background was preliminarily subtracted.
The corresponding description has been added to experimental section.
Comments 13. Gas measurements: info on the chamber volume, on the relative humidity and temperature in the chamber during the exposures, exposure time and recovery time and how the chamber is purged after the exposures should be added. How is the concentration evaluated? Finally, response is defined as (Iair/Igas) at line 214, but in the rest of the manuscript response is defined considering the resistance and not the current.
Response 13. Thank you for your comment. The system is designed to generate vapor flows and measure the gas-sensing properties of materials by monitoring changes in electrical resistance under exposure to target vapors or humidity. The vapor generation setup includes an air pump that supplies air into three separate branches. In the first branch, the airflow is regulated by MFC 1, which works together with MFC 2 and MFC 3 to control the concentration of gas vapors and relative humidity before the flow enters a mixing chamber. The second branch uses MFC 2 to direct air through a bubbler containing the target solution, such as isopropanol, ethanol, or acetone, where the airflow pressure causes the liquid to transition into vapor. Similarly, the third branch generates water vapor by passing air through a bubbler filled with distilled water. These vapor flows are uniformly mixed in a mixing chamber before being introduced into a sample chamber for testing. To calculate the pressure of saturated vapors of the liquid in the bubbler, the Antoine equation was used.
The measurement system consists of the sample chamber (volume 0.193 L), a power supply, control components, and measuring instruments (Figure 12). An Arduino controller manages system parameters, including valve operations and environmental settings, ensuring flexible and precise control. The electrical resistance of the material is measured using a Picoammeter 6485 for small current detection. Measurement data are transferred to a computer for further analysis and processing. This integrated system allows for the precise study of material responses to vapor flow or humidity exposure, making it suitable for evaluating gas-sensing performance.
All measurements have been conducted at ambient temperature of 22-25°C.
Relevant additions have been made to experimental section.
Round 2
Reviewer 1 Report
Comments and Suggestions for Authors
The manuscript was considerably improved after revision.
Accept in the present form.
Author Response
The authors thank the reviewer for a thorough reading of the manuscript and valuable comments.
Reviewer 2 Report
Comments and Suggestions for Authors
The authors partially revised the manuscript taking into account some parts of my previous comments. Nevertheless, a huge issue is present: the sensors are not selective. According to the data presented, there is no way to discriminate between ethanol and isopropanol, at similar concentration, but also the added exposures to acetone are not significative to prove the selectivity. Indeed, 3000 ppm of acetone gives the same response of 500 ppm of ethanol or isopropanol, and this is not a negligible concentration. If the authors cannot provide clear experimental evidence on the selectivity, I still think that the manuscript should not be published.
Additionally, other minor issues still needs to be addressed.
In details, all my comments are as follows:
1) Introduction is still not focused. The authors added only one sentence related to NO2 detection, but it is still not clear the role of the present work for isopropanol and ethanol detection compared to literature. Additionally, it is not mentioned why ethanol and 2-propanol should be detected.
2) Previous comment n3: the reviewer still thinks that the S 2p spectra should be acquired with a better signal-to-noise ratio. Indeed, it is not completely true what the authors wrote: “The peak with an energy of 166.5 eV may be associated with the contribution of sulfur bound to oxygen [http://dx.doi.org/10.17576/jsm-2018-4708-33], but there is no corresponding contribution on the spectrum of the core oxygen level, so it was also not taken into account.”. The contribution of sulfur-oxygen bonds in the O 1s core level spectrum can be found around 530 eV (see for instance: Herrmann, I., Kramm, U. I., Radnik, J., Fiechter, S., & Bogdanoff, P. (2009). Influence of sulfur on the pyrolysis of CoTMPP as electrocatalyst for the oxygen reduction reaction. Journal of The Electrochemical Society, 156(10), B1283.) and from the data presented in panel 3-c, it cannot be excluded the presence of this bond. Therefore, it is mandatory to acquire the S 2p core level spectrum with a proper signal-to-noise ratio.
3) Previous comment n4: authors did not explain how they evaluated response and recovery time (i.e. is it the time required to achieve 80% or 90% or 100% of response/recovery?) Additionally, as already required, the fluctuation of the signal during the exposure should be justified and discussed. Indeed, during the exposure for instance isopropanol, the response changes between 1.24 and 1.30 during to what it is supposed being the same concentration of gas. This change is not negligible.
4) New Figure 7 and Figure 6 (and previous comment n7): up to at least 1500 ppm, it is clear that exposures to the same concentration of ethanol or isopropanol lead to the same response. For instance, considering a response of about 1.5, it can be related to an exposure to 1500 ppm of ethanol or to 2000 ppm of isopropanol. Additionally, 3000 ppm of acetone gives a response of 1.15, which according to the fitting of the calibration curves can be also the response for exposures to about 500 ppm of ethanol and isopropanol.
Selectivity is not proven, and this is a serious issue. How did the authors plan to discriminate between ethanol and isopropanol, if the response is the same? Moreover, the data presented for acetone exposures do not prove selectivity as well. It is true that in that case the concentration is different, still we are dealing with concentrations that can be easily found in a real environment.
Finally, also response and recovery time are quite similar, as well as the shape of the curve; there is no a discriminating parameter between the two gases.
5) New Figure 6 (and previous comment n6): the proposed interpolation does not fit the experimental data. It is clear that there is not a match between the data and fit curve for concentration below 1000 ppm. Please, correct the inconsistency. The detection limit has not been evaluated.
6) Table 1: authors wrote that the response to 1000 ppm of ethanol is 1.38, in table 1, while in the manuscript (line 161, page 6) they reported 1.35; please correct the inconsistency.
7) Previous comment n9 and new figure S1: the reported cycles do not prove the reproducibility. Indeed, a complete recovery is not achieved, meaning that at least the recovery time is not repeatable. Additionally, the response of the first cycle is about 1.30, while the second cycle is around 1.43, therefore also the response is not completely reproducible.
8) Previous comment n10: referee still thinks that exposures to at least ammonia (which can be exploited in liquid form as required by the experimental set up) should be performed. Also in light to the fact that there is no selectivity for alcohols, it is better to test a different analyte, and ammonia is a good candidate. Ammonia is usually found and tested in concentration ranges from low ppm up to hundred ppm. Please consider adding those exposures.
9) Previous comment n10: authors exposed the sensors to different RH value. It is clear that the sensors are highly influenced by the humidity conditions. As already required, but now that there is the proof that RH strongly changes the responses, exposures at different RH values for ethanol and isopropanol should be performed.
10) Consider Figure 5 and Figure 8: can the authors explain while the curve presented in Fig 8 shows a complete recovery and a response without fluctuations, whereas the curves reported in Fig 5 presents a huge fluctuation in response (see also comment n3) and the recovery is not completely achieved? It does not seem related to the used set up, since it is the same.
11) References are not in the correct format.
Author Response
We would like to thank the reviewer for his careful and detailed report. We appreciate his comments and suggestions and have made every effort to address them in the revised manuscript. The changes made in the revised version are clearly indicated.
The lack of selectivity is a fundamental property of semiconductor adsorption sensors. The change in conductivity is fundamentally different when exposed to oxidizing and reducing gases. All reducing gases interacting with oxygen adsorbed on the surface of the n-type semiconductor lead to an increase in conductivity (decrease in resistance). Attempts to increase selectivity are limited to the choice of temperature conditions, doping and other techniques.
The main development of adsorption semiconductor sensors occurred in the last century [Madou MJ, Morrison SR (1989) Chemical Sensing with Solid State Devices, Academic Press Inc., Boston].
However, many opportunities were missed, leading to a lack of further development in this area. The situation began to change when industrial development led to the use of modern equipment for developing microelectronic sensors [https://doi.org/10.1007/s002160051490]. At the same time, an approach based on receiving signals from multiple sensors simultaneously, i.e., using multisensor systems, was proposed in Karlsruhe [https://doi.org/10.1021/nl071815.]. Currently, the most common platform contains 38 sensitive elements. However, there are exceptions. For example, tin dioxide doped with copper provides a selective response to hydrogen sulfide [https://doi.org/10.1016/j.snb.2019.127179 ].
Comments 1. Introduction is still not focused. The authors added only one sentence related to NO2 detection, but it is still not clear the role of the present work for isopropanol and ethanol detection compared to literature. Additionally, it is not mentioned why ethanol and 2-propanol should be detected.
Response 1. The following text has been added to the introduction to highlight the problem of alcohol sensors and current work in this area.
Volatile organic compounds released into the atmosphere as a result of industrial production pose a threat to the environment and human health. Among them, isopropyl alcohol C3H8O, used in medicine, chemical and other industries, has narcotic and toxic effects on humans. Ethanol detection is of interest for assessing the condition of drivers in order to ensure safety on the roads, as well as in the food industry.
A CuO-based sensor doped with Sn was proposed to assess the ethanol content in the air exhaled by the driver, the optimal operating temperature of which is 200 °C [57]. Gold-decorated SnO2 nanotubes can detect to ethanol at 160 °C [58]. The yolk-shell Bi2MoO6 gas sensor showed the response to isopropanol at 270°C [59]. MoO3 nanoflakes have also been developed, for which the optimal detection temperature for ethanol is 350 °C and isopropanol is 200°C [60]. In a number of studies, sensors based on tungsten oxide have been obtained, demonstrating high operating temperatures for response to alcohols, as shown in [61-63]. However, the problem of high operating temperatures in alcohol sensors has not yet been fully addressed in most work in this field.
Comments 2. Previous comment n3: the reviewer still thinks that the S 2p spectra should be acquired with a better signal-to-noise ratio. Indeed, it is not completely true what the authors wrote: “The peak with an energy of 166.5 eV may be associated with the contribution of sulfur bound to oxygen [http://dx.doi.org/10.17576/jsm-2018-4708-33], but there is no corresponding contribution on the spectrum of the core oxygen level, so it was also not taken into account.”. The contribution of sulfur-oxygen bonds in the O 1s core level spectrum can be found around 530 eV (see for instance: Herrmann, I., Kramm, U. I., Radnik, J., Fiechter, S., & Bogdanoff, P. (2009).. Journal of The Electrochemical Society, 156(10), B1283.) and from the data presented in panel 3-c, it cannot be excluded the presence of this bond. Therefore, it is mandatory to acquire the S 2p core level spectrum with a proper signal-to-noise ratio.
Response 2. Fig. 3,d is replaced. The S2p spectrum with a lower noise level is presented. Deconvolution of the spectrum with subsequent analysis showed that it consists of two peaks. The predominant peak with a binding energy of 163.8 eV corresponds to the S2p 1/2 line in the WS2 compound [67, 69]. The minor component may be related to the oxidized state of sulfur [70].
Comments 3. Previous comment n4: authors did not explain how they evaluated response and recovery time (i.e. is it the time required to achieve 80% or 90% or 100% of response/recovery?) Additionally, as already required, the fluctuation of the signal during the exposure should be justified and discussed. Indeed, during the exposure for instance isopropanol, the response changes between 1.24 and 1.30 during to what it is supposed being the same concentration of gas. This change is not negligible.
Response 3. The calculation of response and recovery time is described in lines 349-350 on page 11.
Comments 4. New Figure 7 and Figure 6 (and previous comment n7): up to at least 1500 ppm, it is clear that exposures to the same concentration of ethanol or isopropanol lead to the same response. For instance, considering a response of about 1.5, it can be related to an exposure to 1500 ppm of ethanol or to 2000 ppm of isopropanol. Additionally, 3000 ppm of acetone gives a response of 1.15, which according to the fitting of the calibration curves can be also the response for exposures to about 500 ppm of ethanol and isopropanol.
Selectivity is not proven, and this is a serious issue. How did the authors plan to discriminate between ethanol and isopropanol, if the response is the same? Moreover, the data presented for acetone exposures do not prove selectivity as well. It is true that in that case the concentration is different, still we are dealing with concentrations that can be easily found in a real environment.
Finally, also response and recovery time are quite similar, as well as the shape of the curve; there is no a discriminating parameter between the two gases.
Response 4. The developed sensor shows similar responses to alcohols and a slightly lower response to acetone. To ensure selectivity, the developed sensor layers can be used to create multisensor systems.
Comments 5. New Figure 6 (and previous comment n6): the proposed interpolation does not fit the experimental data. It is clear that there is not a match between the data and fit curve for concentration below 1000 ppm. Please, correct the inconsistency. The detection limit has not been evaluated.
Response 5. Figure 6 has been replaced by an approximation for the concentration range between 1000 and 3000 ppm. A detection limit has been established for all analyzed gases.
The following additional text has been included in the manuscript:
In the section describing the experiment: The detection limit was estimated as 3[C]/((Rair-Rgas)/σ),whereCis the concentration of the targetgas andσis the signalfluctuation[DOI:10.1039/d2ra07607a].
In the results and discussion:
The calculated detection limits are 500 ppm for ethanol, 600 ppm for isopropanol and 750 ppm for acetone.
Comments 6. Table 1: authors wrote that the response to 1000 ppm of ethanol is 1.38, in table 1, while in the manuscript (line 161, page 6) they reported 1.35; please correct the inconsistency.
Response 6. The inconsistency is corrected.
Comments 7. Previous comment n9 and new figure S1: the reported cycles do not prove the reproducibility. Indeed, a complete recovery is not achieved, meaning that at least the recovery time is not repeatable. Additionally, the response of the first cycle is about 1.30, while the second cycle is around 1.43, therefore also the response is not completely reproducible.
Response 7. Indiscussion of the results(line188-199,page6) the variation of responseandrecovery is described.This is dueto the peculiarities of the interaction of the sensorlayerwith the targetgasesatroomtemperature.Forrecovery, alongerpurge of the sample with driedair is necessary.
The following text has been added to the discussion of the results: For recovery, a longer purge of the sample with dried air is necessary.
Comments 8. Previous comment n10: referee still thinks that exposures to at least ammonia (which can be exploited in liquid form as required by the experimental set up) should be performed. Also in light to the fact that there is no selectivity for alcohols, it is better to test a different analyte, and ammonia is a good candidate. Ammonia is usually found and tested in concentration ranges from low ppm up to hundred ppm. Please consider adding those exposures.
Response 8. Experiments have been conducted to investigate the response of the nanocomposites to ammonia in a concentration range of 800-2400 ppm. The results are presented in Fig. S4. The response to 1600 ppm of NH3 was 1.1, which is significantly lower than the corresponding values for volatile organic compounds. Additionally, incomplete recovery was observed within 2 minutes, due to the inability of rapid desorption of reaction products at room temperature.
Comments 9. Previous comment n10: authors exposed the sensors to different RH value. It is clear that the sensors are highly influenced by the humidity conditions. As already required, but now that there is the proof that RH strongly changes the responses, exposures at different RH values for ethanol and isopropanol should be performed.
Response 9. These results are presented in the revised version of the article after the first round of review. The response to isopropyl alcohol under conditions of varying humidity was shown in Fig. S5 as a function of resistance versus time. Figure 9 also shows the responses to all the studied gases with different concentrations with varying humidity. This figure shows a decrease in response with an increase in relative humidity.
Comments 10. Consider Figure 5 and Figure 8: can the authors explain while the curve presented in Fig 8 shows a complete recovery and a response without fluctuations, whereas the curves reported in Fig 5 presents a huge fluctuation in response (see also comment n3) and the recovery is not completely achieved? It does not seem related to the used set up, since it is the same.
Response 10. Figure 8 shows the interaction with water vapor and Figure 5 shows the interaction with volatile organic compounds. Due to the different nature of these substances, the products of the reaction are not always easily removed when they interact with volatile organic compounds. In contrast, water adsorption is reversible.
Comments 11. References are not in the correct format.
Response 11. The references format has been corrected.
Round 3
Reviewer 2 Report
Comments and Suggestions for Authors
Authors revised the manuscript, but the selectivity issue is not yet properly discussed and commented. According to the reply in comment n4, they plan to use the sensor in an array, instead of as single sensor, due to the lack of selectivity. This may be a solution. Nevertheless, this aspect is not properly discussed in the manuscript. Please, comment and introduce this possibility.